# Modular Refinement of Small Language Models for Physics Reasoning via Localized Error Feedback

## Abstract

Large Language Models (LLMs) excel at many reasoning tasks but struggle with scientific domains like physics, which demand precise mathematical calculations alongside deep conceptual and factual understanding. In complex physics problem-solving, LLMs commonly falter due to three core issues: misunderstanding the problem, incorrect application of concepts, and calculation mistakes. These challenges are more pronounced in small LLMs due to their limited capacity, making them more prone to failures. To address these limitations, we propose a modular reinforcement learning refinement framework tailored for small LLMs, integrating first-step error localization and correction through a Reinforcement Learning agent-guided feedback mechanism. We also introduce PhysicsQA, a diverse benchmark of 370 physics problems designed to evaluate LLM reasoning across the aforementioned dimensions. Using our Framework, experimental results demonstrate improvements up to 16% in final answer accuracy reasoning using Small language models over existing results.

## 1 Introduction

Scientific reasoning, particularly in the field of physics, requires a deep understanding that spans multiple disciplines. It demands not only domain-specific knowledge but also the integration of mathematical calculation with theoretical concepts, applying abstract principles and formulae across various contexts and scenarios. Unlike purely mathematical reasoning, which is abstract and symbolic, physics reasoning requires grounding those abstractions in real-world phenomena and physical laws. Successfully solving these challenges is a fundamental aspect of human intelligence, as it entails not just recalling information but adapting knowledge to solve diverse complex problems. Large language models (LLMs) are growing in size, but bigger is not always better Boye & Moell (2025). Solving complex reasoning problems still an open challenge for small open source LLMs Srivastava et al. (2025) models.

Chain of Thoughts (CoT) Wei et al. (2022b) can enable LLMs to solve complex reasoning tasks, it is highly sensitive to individual mistakes and vulnerable to error accumulation Shen et al. (2021). If a tiny mistake occurs, it can change the meaning of the whole statement Xiao et al. (2023), leading to incorrect answers Cobbe et al. (2021). Alternatively Retrieval-Augmented Generation (RAG) improves factual accuracy by incorporating external knowledge, but they primarily focus on information retrieval rather than reasoning correction. One approach to address these challenge can be collecting question and solution trajectory annotations and finetune LLMs to enhance these capabilities, similar to recent mathematical reasoning works Luo et al. (2023); Yuan et al. (2024). However, the process of such annotations and finetuning is time-consuming and costly. Finetuning may sometimes lead to a larger decline in CoT reasoning performance and can compromise the faithfulness of CoT reasoning Lobo et al. (2024). It also leads to challenges such as catastrophic forgetting McCloskey & Cohen (1989). Reinforcement Learning from Human Feedback (RLHF) is perhaps the most well-known application of RL techniques for finetuning LLMs. Reward models in RLHF often favor verbosity, leading models to generate unnecessarily long or redundant outputs Chiang & Lee (2024).However, despite these advances, RL-enhanced LLMs still rely primarily on internal knowledge and language modeling Wang et al. (2024). This reliance becomes a major limitation for time-sensitive or knowledge-intensive questions, where the model's static knowledge

base may be outdated or incomplete, often resulting in inaccuracies or hallucinations . Agentic reasoning addresses these limitations by enabling LLMs to dynamically interact with both external resources and environments throughout the reasoning process Xiong et al. (2025); Patil & Jadon (2025),Open source small LLMs (SLMs) struggles to directly identify reasoning mistakes in their own solutions Li et al. (2024); Tyen et al. (2024), making them unreliable for self verification and refinement. SLMs lack the necessary depth Zhang et al. (2024) to reliably detect mistakes or rectify their reasoning processes independently.

Based on the limitations of above Techniques to improved and correct Scientific reasoning as mentioned, our key contributions are:

- PhysicsQA Benchmark: Introduced PhysicsQA, a dataset of 370 diverse, intermediate-level high school physics problems with verified chain-of-thought solutions to evaluate LLM reasoning.
- Error Taxonomy: Identified three core LLM error types in physics problem solving: Problem Miscomprehension, Incorrect Concept Application, and Calculation Errors.
- Agent Guided Feedback Training: Designed a modular training pipeline using specialized agents to provide localized, error-specific feedback based on the identified error types.
- Reinforcement Learning Framework: Built a reinforcement learning framework for small LLMs that leverages error localization and structured feedback to improve physics reasoning.
- Empirical Gains: Achieved substantial accuracy improvements on multiple benchmarks, outperforming existing methods for small open-source LLMs.

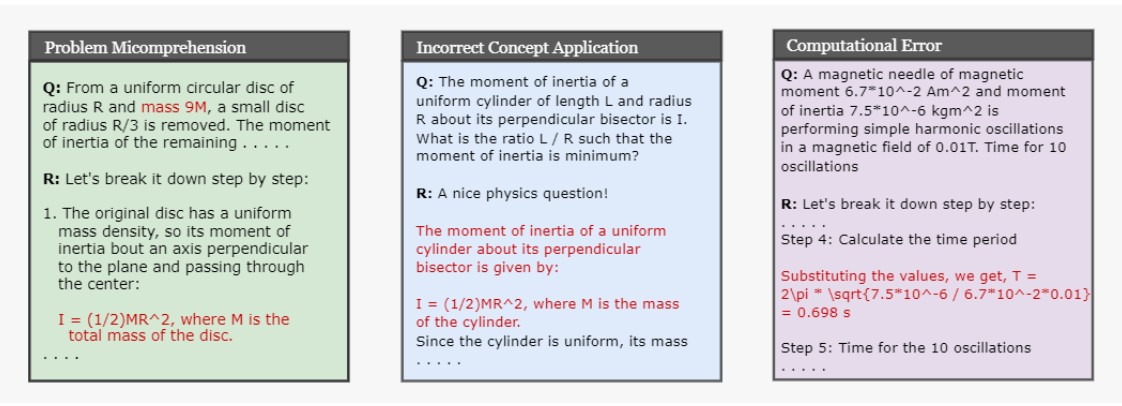

Figure 1: The illustration of three key error observations in the CoT solution of SLMs for physics problems (highlighted in red). (a) showcases problem miscomprehension, where the SLM response uses the incorrect value of variables given in the question here, M instead of 9M, (b) showcases incorrect concept application in the SLM response, here incorrect moment of inertia formula for uniform cylinder, (c) demonstrate Calculation error within SLM response here, incorrect calculation of time period.

## 2 RELATED WORKS

### 2.0.1 LLM REASONING FOR PHYSICS

Researchers have begun exploring the potential of LLMs as reasoning tools in the physics domain Anand et al. (2024); Ding et al. (2023); Pang et al. (2024). Studies have demonstrated that LLMs can solve complex word problems requiring calculation and inference, often achieving near human-level accuracy, especially with effective prompting techniques such as few-shot learning using similar examples Ding et al. (2023), leveraging RLHF Anand et al. (2024) or implementing agentic system Pang et al. (2024). While much of this research focuses on general physics reasoning. LLMs have

been successfully applied to address multi-step reasoning tasks by generating intermediate reasoning steps, referred to as CoT, Auto-CoT Zhang et al. (2022), and Complex-CoT Fu et al. (2022), among others. LLMs tends to struggle with arithmetic calculations when solving math problems Gao et al. (2023), Leveraging the strengths of GPT4 Code Interpreter Zhou et al. (2023) has been integral to frameworks like MathCoder Wang et al. (2023), which is designed to improve the mathematical calculations capabilities of open-source models.

### 2.0.2 LLM Reasoning with Fine tuning and RLHF

While large LLMs exhibit excellent performance on mathematical reasoning tasks, adapting smaller models for these tasks remains an open problem Wei et al. (2022a). Some methods used questions from existing training datasets and used prompting to generate solutions for fine tuning (FT) smaller models Ho et al. (2022); Fu et al. (2023). Others use various techniques to rephrase the questions to create more examples Vu et al. (2020) or multiple views of solutions Liang et al. (2023) to achieve better reasoning performance.Reinforcement Learning from Human Feedback (RLHF) Ouyang et al. (2022); Glaese et al. (2022) most often works by training a reward model to capture human preferences over a task. Full Step DPO Xu et al. (2025), a novel framework for mathematical reasoning that optimizes each step in the entire reasoning chain using step wise rewards.

### 2.0.3 LLM Reasoning with external database

Lewis et al. (2020) proposed Retrieval-Augmented Generation (RAG) framework, which incorporates a retrieval component to fetch relevant information from a given knowledge base. Integrating LLMs with knowledge representation tools, such as knowledge graphs (KGs) Mruthyunjaya et al. (2023), has further enhanced reasoning capabilities. Yao et al. (2024) demonstrated that augmenting LLMs with comprehensive external knowledge from KGs can significantly improve their performance and facilitate more robust reasoning processes.

### 2.0.4 Self Verification with LLMs

Recent works Cobbe et al. (2021); Ling et al. (2024) have attempted to address the challenge of error detection in step-by-step reasoning. Miao et al. (2023) proposes using the LLM itself to verify the conditional correctness of each step in the reasoning chain, similar to how a human reviews their work. Accurate error recognition and correction are crucial for enhancing problem solving capabilities, as demonstrated by Li et al. (2024), which defines tasks to assess LLMs' mathematical reasoning abilities in error identification and correction.

## 3 Dataset : PhysicsQA

Benchmarks like MMLU Hendrycks et al. (2020), SciEval Sun et al. (2024), focus on foundational knowledge, while more challenging ones like OlympiadBench He et al. (2024) and JEEBench Arora et al. (2023) require advanced reasoning skills. To bridge the gap, we curated our own dataset PhysicsQA, containing the set of diverse, intermediate-level high school physics problems that provide a balanced challenge, allowing a exhaustive evaluation and analysis of LLMs on physics problems. It has 370 high-school Indian Engineering Exam (JEE) physics problems sourced from PW Live (2024); Doubtnut that offer diversity in both topic coverage and difficulty. These problems are notably challenging, often requiring the application of multiple concepts, intricate calculations, and multihop reasoning. Each problem is accompanied by a correct CoT solution, verified using GPT4 with human annotators in the loop. This enables researchers to assess physics reasoning beyond final answer accuracy and conduct rigorous, step by step evaluations of LLMs conceptual understanding and calculation abilities. Constructions details and chapters breakdown illustrated in 3

## 4 Methodology

In this section, we present our Agent-Guided Reinforcement Learning framework for enhancing physics reasoning for SLMs. Our approach begins with a taxonomy of fundamental error types commonly observed in complex physics problem solving. Building on this foundation, As illustrated in Figure 2, we introduce a structured RL training pipeline composed of three core components: (1)

an error localization model that identifies the earliest mistake in the model's reasoning; (2) An agentic feedback generator that produces localized, error-specific guidance; and (3) a reinforcement learning loop that fine-tunes the model to revise its reasoning using the provided feedback. In the following subsections, we detail each of these components and how they interact to enable effective physics reasoning in SLMs.

## 4.1 TRAINING DATASET

We curated a finetuning dataset comprising 2,494 high school-level physics questions sourced from standard materials Pandey; Tipler (1999); Pinsky (1989); Halliday et al.which is different source from our benchmark PhysicsQA. Each question is paired with a detailed CoT solution. The dataset provides a diverse and challenging set of physics problems, accompanied by oracle level step by step reasoning, following the same process used for PhysicsQA. Training Sample shown in fig.19

## 4.2 LoRA SLM TRAINING WITH WARMUP

To make the RL training effective, we first warm-start the SLM through supervised finetuning, enabling it to internalize core reasoning structures and domain-specific solution patterns as shown in fig 2. For the subsequent reinforcement learning stage, we adopt LoRA adapters, which substantially reduce memory and compute overhead while training with our objective. This combination of warm initialization and parameter-efficient finetuning enables our framework to teach strong physics reasoning capabilities to small LLMs without full model finetuning.

## 4.3 PHYSICS ERROR TAXONOMY

Based on a systematic analysis of solutions generated by wide range LLMs for complex physics problems, we identify three recurring categories of physics reasoning errors, as shown in Fig.1.

**Problem Miscomprehension**, SLMs in few cases struggle to fully grasp the objective of the question, along with misinterpreting the values of variables and constants provided in the question. These misinterpretations result in fundamentally flawed reasoning trajectories that do not address the intended problem. **Conceptual Misapplication**, SLMs struggle to apply the correct concepts or formulae with respect to the context of the given problem. This issue is a more recurring one in SLMs, especially for problems requiring considering a specific case rather than relying on a generic formula. **Calculation Errors**, Many physics problems involve mathematical reasoning and algebraic calculation, areas where SLMs tend to struggle. They often make mistakes in these calculations, which propagate through the solution and affect both intermediate reasoning and final answers.

## 4.4 ERROR LOCALIZATION

For error localization, we utilize LLaMa 3.1 405B. Prior work by Zhang et al. (2024) has shown that GPT4o can effectively use gold labels from initial solutions as signals for self-refinement. This oracle verifier setup serves as an upper bound on the performance of refinement agents. Inspired from this framework, we prompt LLaMa 3.1 405B to identify and localize errors in the solution, along with reasoning about the cause of each error. Our prompt based verifier performs three tasks: (a) Identify the first error step in the model-generated solution where model deviates from the correct reasoning.(b) Classify the error into one of three categories:(c) Problem Miscomprehension, Conceptual Error, or Calculation Error. Explain the error briefly, describing what went wrong in the first error reasoning step.

We utilize LLaMa 3.1 405B solely for error identification as shown Figure 2, which is used in our reward modeling and informs the routing of the error step to the appropriate feedback agent. It helps each agent to understand the initial point of failure, enabling effective and focused feedback for that stage.

## 4.5 AGENTIC FEEDBACK GENERATION

We developed our feedback generator as shown in Figure 2, which produces structured, step-level refinement feedback to guide SLMs in correcting localized reasoning errors. It consists of a set of

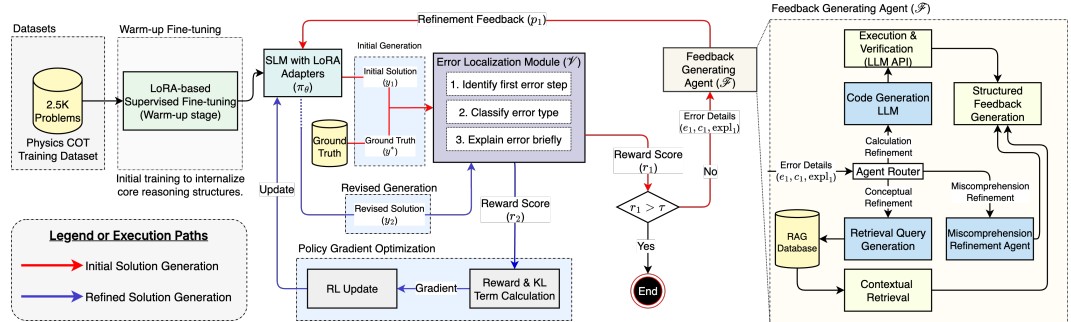

Figure 2: The architecture of our Agent-Guided Reinforcement Learning (AG-RL) framework. The process begins with a supervised fine-tuning warm-up stage. Subsequently, the SLM enters the iterative RL loop, starting with the Initial Policy Execution (red arrow) to generate a preliminary solution ($y_1$). This solution is evaluated by the Feedback Generating Agent (F), which localizes the first error and generates structured feedback. This feedback informs a policy gradient update, triggering the Updated Policy Execution (blue arrow) to produce a revised solution ($y_2$). A reward score, calculated from this revised solution, drives the final policy update.

specialized refinement agents, each designed to address one of the three core challenges in physics reasoning. The generated feedback is then incorporated into the model's reasoning process to revise and improve its original solution attempt.

Miscomprehension identified in SLM generated solutions, can be corrected through instruction prompting. The agent handles such cases by incorporating the provided error explanation into a targeted feedback prompt, instructing the model to revise its understanding of the problem and generate a fully corrected solution in the next attempt.

To address incorrect concepts and misapplied formulae in the identified error step, we leverage RAG to produce targeted, context-aware corrections. Given a physics question, an erroneous reasoning step, and an explanation of the conceptual mistake, the agent performs the following sequence:

**Retrieval Query Generation**, The agent generates a focused query using the question, error step, and explanation. This query is optimized to retrieve the most relevant conceptual content from a domain-specific corpus. **Contextual Retrieval**, Using a vector store built with sentence-transformer embeddings, the agent retrieves key conceptual content required to correct the identified error step. **Structured Feedback Generation**, The retrieved context and original error description are used to generate structured, corrective feedback. This feedback is designed to help the model understand its mistake and apply the necessary correction in its reasoning.

We adopt a code generation approach to refine calculation and mathematical errors in the identified errorneous step. By leveraging executable code as a verification mechanism, the agent ensures accurate calculation correction and interpretability in the reasoning. Given a physics question, an erroneous step, and an explanation of the mistake, that guides small LLMs toward step-level correction. The agent follows a three-step process: **Code Generation**, The agent generates Python code that correctly performs the intended calculation, including proper variable initialization, unit conversions, and mathematical logic. **Execution & Verification**, The generated code is executed to validate correctness and produce a definitive numerical result. **Feedback Generation**, Based on the code and its output, the agent generates natural language feedback that explains the correct logic, clarifies the original error, and ensures unit consistency.

### 4.6 Agent Guided RL Training

Self Correction remains a highly desirable yet largely ineffective capability in current LLMs Kamoi et al. (2024). To address this Kumar et al. (2024) introduced SCoRe, a multi-turn reinforcement learning (RL) framework aimed at teaching models to revise their own mistakes. This enables the model to explore diverse reasoning trajectories and potentially converge on more robust solutions.

However, RL based training pipelines are often complex and computationally demanding. Sidahmed et al. (2024); Wang et al. (2025) demonstrate that integrating LoRA Hu et al. (2022) into RL setups allows for efficient adaptation of reasoning patterns while retaining core model knowledge significantly reducing training time, memory usage, and overall cost.

Building on these foundations, we introduce an efficient agent-guided RL training pipeline as shown in figure 2, that improves physics reasoning in SLMs by refining their CoT reasoning using localized error and agentic feedbacks.

Given a physics problem $x$, the model generate a first attempt solution $y_1 \sim \pi_\theta(\cdot \mid x)$. This solution is passed to the error localization model, which identifies the first erroneous step in the model reasoning, categorizes the type of error (miscomprehension, conceptual, or calculation), and provides a brief explanation. If the model's first response is deemed sufficiently accurate quantified by a reward score $r_1 > 0.95$, the attempt is considered correct and skipped from the correction loop. Otherwise, the feedback-generating agent uses the localized error information to construct a correction feedback prompt $p_1$, which is given as the input to the model to produce a revised attempt $y_2 \sim \pi_\theta(\cdot \mid x, y_1, p_1)$. The agentic feedback assist the SLM to explore correct and robust reasoning trajectory in the revised attempt $y_2$.

### 4.7 REWARD DESIGN

The reward function is designed to encourage SLM to progressively refine the first error step in its reasoning and is defined as:

$$r = \frac{e_{\text{first}}}{n + 1}$$

where $e_{\text{first}}$ is the first eror step number and n is the total number of steps in the generated solution. The first error step here is identified using the error localization module. This reward signal penalizes earlier mistakes and encourages the model to push the first error further along the reasoning chain. As training progresses, the model learns to push and ultimately eliminate its first error, thereby producing increasingly accurate step-by-step solutions.

Given the revised attempt $y_2$, we evaluate it using our reward function resulting in the reward score $r_2$. The training objective is to shift the first error further down the reasoning chain or completely eliminate it in $y_2$ using agentic feedback and is given by:

$$\max_\theta \mathbb{E}_{x, y_1, y_2} \left[ \hat{r_2}(y_2, y^*) - \beta_2 D_{\text{KL}} \left( \pi_\theta(\cdot \mid x) \parallel \pi_{\text{ref}}(\cdot \mid x) \right) \right]$$

Here, $r_2$ reflects how much the model response has improved after agentic feedback. A higher $r_2$ indicates a successful refinement, such as correcting an earlier mistake or pushing the first error further in the reasoning chain. The KL regularization term ensures the updated policy remains close to a reference policy during training, promoting stable optimization. This setup allows the model to iteratively improve its reasoning capabilities over training steps, guided by localized, high-quality agentic feedback.

### 4.8 TRAINING SETUP

The warmup stage finetuning was done for 2 epoch using a single H100 GPU using AdamW optimizer and a learning rate of 5e-5 and a batch size of 4. For agentic RL training, we utilized Distributed Data Parallel in PyTorch, using AdamW optimizer, learning rate of 5e-6, and a batch size of 4 per GPU. The RL training was performed for 6 epochs using 4 H100 GPUs.

## 5 EXPERIMENTS

### 5.0.1 BENCHMARK DATASETS

In our experiments, we used four datasets: SciEval-Static, MMLU High School and College, JEEBench and PhysicsQA. SciEval-Static is a subset of SciEVal , consisting 164 questions from physics divided into multiple sub-topics. MMLU, consists of a 118 College level and 173 high

| LLM Models | SciEval-Static | | | MMLU-High | | | MMLU-College | | | JEEBench | | | PhysicsQA | | |
|---|---|---|---|---|---|---|---|---|---|---|---|---|---|---|---|
| | AO | CoT | 3-Shot | AO | CoT | 3-Shot | AO | CoT | 3-Shot | AO | CoT | 3-Shot | AO | CoT | 3-Shot |
| **Ultra Small** | | | | | | | | | | | | | | | |
| Qwen 2.5 1.5B Instruct | 50.60 | 62.73 | 53.65 | 32.94 | 47.05 | 31.12 | 45.45 | 52.62 | 42.23 | 46.34 | 37.53 | 41.46 | 27.29 | 30.64 | 34.59 |
| LLaMa 3.2 1B Instruct | 40.24 | 44.51 | 35.97 | 30.00 | 30.58 | 32.34 | 26.36 | 33.63 | 29.32 | 30.89 | 32.52 | 34.95 | 20.27 | 21.35 | 22.16 |
| LLaMa 3.2 3B Instruct | 45.12 | 62.26 | 48.17 | 28.23 | 50 | 44.32 | 32.72 | 53.63 | 43.65 | 43.90 | 30.21 | 40.65 | 27.02 | 27.67 | 40.81 |
| Phi 3.5 mini 3.8B Instruct | 54.87 | 62.43 | 62.19 | 48.23 | 55.17 | 54.67 | 41.81 | 65.63 | 60.50 | 43.08 | 35.90 | 43.08 | 32.70 | 33.35 | 33.51 |
| **Small** | | | | | | | | | | | | | | | |
| LLaMa 3.1 8B Instruct | 48.17 | 79.26 | 45.73 | 43.52 | 57.64 | 51.26 | 40.90 | 61.81 | 63.35 | 39.83 | 38.21 | 39.83 | 27.02 | 35.67 | 33.24 |
| OLMo 7B Instruct-hf | 42.68 | 39.63 | 44.51 | 26.47 | 28.82 | 29.83 | 36.36 | 28.18 | 35.26 | 34.14 | 32.52 | 31.23 | 12.97 | 22.16 | 21.89 |
| Phi 3 medium 14B Instruct | 59.14 | 79.87 | 65.85 | 54.11 | 67.64 | 58.75 | 52.72 | 80.90 | 68.37 | 47.15 | 39.02 | 39.02 | 33.51 | 51.35 | 54.32 |
| **Intermediate** | | | | | | | | | | | | | | | |
| Qwen2.5 32B Instruct | 78.04 | 90.85 | 81.70 | 68.82 | 89.41 | 83.25 | 63.63 | 91.81 | 83.25 | 51.65 | 57.72 | 60.16 | 53.24 | 75.20 | 78.08 |
| Qwen 2.5 72B Instruct | 71.95 | 98.29 | 76.82 | 71.76 | 88.23 | 75.63 | 67.27 | 90.00 | 84.56 | 50.40 | 62.60 | 58.53 | 54.59 | 75.94 | 81.08 |
| LLaMa 3 70B Instruct | 70.73 | 91.46 | 87.80 | 62.35 | 85.29 | 78.35 | 60.90 | 90.90 | 87.89 | 53.65 | 58.53 | 65.85 | 43.78 | 76.21 | 74.05 |
| Gemma 2 27b Instruct | 56.09 | 82.92 | 61.58 | 53.52 | 70.00 | 69.18 | 56.36 | 77.27 | 78.29 | 45.52 | 47.96 | 44.71 | 37.02 | 52.97 | 63.78 |
| **Large** | | | | | | | | | | | | | | | |
| LLaMa 3.1 405B Instruct | 80.48 | 87.80 | 84.14 | 73.52 | 85.29 | 79.36 | 78.00 | 87.27 | 85.43 | 62.50 | 68.33 | 70.00 | 51.08 | 74.32 | 70.00 |
| Mixtral 8x7B Instruct v0.1 | 53.25 | 64.37 | 65.62 | 46.47 | 61.17 | 62.87 | 49.09 | 53.63 | 52.31 | 29.16 | 30.00 | 37.50 | 34.32 | 38.91 | 43.78 |
| Mixtral 8x22B Instruct v0.1 | 59.37 | 64.37 | 65.62 | 51.76 | 72.94 | 73.65 | 46.36 | 80.00 | 65.43 | 45.83 | 47.50 | 48.33 | 37.02 | 55.94 | 59.72 |
| Deepseek R1 685B | 53.25 | 61.28 | 65.38 | 50.12 | 58.35 | 60.12 | 45.34 | 55.25 | 56.12 | 36.13 | 44.35 | 51.65 | 39.13 | 44.28 | 43.15 |
| **Proprietary** | | | | | | | | | | | | | | | |
| GPT 4o | 64.02 | 92.68 | 81.09 | 62.71 | 94.06 | 87.28 | 70.00 | 84.70 | 84.17 | 51.67 | 66.67 | 60.83 | 49.45 | 79.45 | 78.37 |
| Gemini 1.5 Flash | 68.29 | 85.97 | 81.70 | 58.47 | 79.66 | 80.05 | 60.58 | 72.35 | 72.94 | 40.00 | 61.66 | 59.87 | 44.86 | 62.97 | 69.72 |

Table 1: We report final answer accuracy (%) of different prompting strategies Answer-Only (AO), Few-shot (3 shot), and Chain of Thought (CoT) across five Evaluation Physics benchmark Dataset. Models span a wide range of scales and architectures, including instruction-tuned LLM model variants. Accuracy highlighted in green indicates best performance.

| Ultra Small | SciEval-Static | | | | | MMLU-High | | | | | MMLU-College | | | | |
|---|---|---|---|---|---|---|---|---|---|---|---|---|---|---|---|
| | CoT | RAG | FT | DPO | Ours | CoT | RAG | FT | DPO | Ours | CoT | RAG | FT | DPO | Ours |
| Qwen 2.5 1.5B Instruct | 62.73 | 68.12 | 57.93 | 59.32 | 79.29 | 47.05 | 59.41 | 41.18 | 46.47 | 55.03 | 52.62 | 62.27 | 43.22 | 43.22 | 53.64 |
| LLaMa 3.2 1B Instruct | 44.51 | 57.37 | 55.49 | 56.93 | 68.09 | 30.58 | 38.02 | 42.35 | 39.41 | 55.29 | 33.63 | 45.82 | 38.14 | 36.44 | 57.89 |
| LLaMa 3.2 3B Instruct | 62.26 | 66.87 | 61.59 | 57.93 | 81.52 | 50 | 55.29 | 48.29 | 47.65 | 67.20 | 53.63 | 61.82 | 45.76 | 50.85 | 69.09 |
| Phi 3.5 mini 3.8B Instruct | 62.43 | 61.01 | 59.17 | 56.87 | 65.19 | 55.17 | 60.53 | 55.97 | 56.12 | 65.83 | 65.63 | 67.27 | 62.77 | 64.34 | 73.12 |

Table 2: Final answer accuracy (%) across Physics Benchmarks for foundational knowledge using different strategies : Vanilla Chain-of-Thought prompting (CoT), Retrieval-Augmented Generation (RAG), supervised finetuning (FT), Direct Preference Optimization (DPO), and proposed framework (Ours). Accuracy highlighted in green indicates best performance.

school multiple-choice questions from various disciplines. JEEBench consists of 123 questions from Physics. Our PhysicsQA comprises 370 carefully selected complex high school physics questions sourced from online resources.

### 5.0.2 LLMs

We conduct experiments across a diverse set of open and closed-source LLMs, ranging from 1.5B to 405B parameters, to evaluate their physics reasoning capabilities on benchmark. The models, including Qwen2.5 1.5B, 32B and 72B Team (2024), LLaMA 3.1 8B , 70B, 405B, LLaMA 3.2 1B, 3B and LLaMA 3.3 70B Grattafiori et al. (2024), Phi 3.5 3.8B mini and Phi 3 14B medium Microsoft

| Ultra Small | JEEBench | | | | | PhysicsQA | | | | |
|---|---|---|---|---|---|---|---|---|---|---|
| | CoT | RAG | FT | DPO | Ours | CoT | RAG | FT | DPO | Ours |
| Qwen 2.5 1.5B Instruct | 37.53 | 40 | 39.02 | 30.08 | 54.86 | 30.64 | 36.91 | 23.51 | 24.32 | 49.62 |
| LLaMa 3.2 1B Instruct | 32.52 | 35.50 | 28.46 | 35.77 | 49.17 | 21.35 | 28.64 | 23.24 | 18.92 | 39.02 |
| LLaMa 3.2 3B Instruct | 30.21 | 40.83 | 36.59 | 38.21 | 57.34 | 27.67 | 31.21 | 27.84 | 25.95 | 46.73 |
| Phi 3.5 mini 3.8B Instruct | 35.90 | 41.25 | 38.04 | 41.65 | 50.23 | 33.35 | 41.59 | 39.19 | 41.49 | 53.22 |

Table 3: Final answer accuracy (%) across Complex Physics Reasoning Benchmarks using different strategies : Vanilla Chain-of-Thought prompting (CoT), Retrieval-Augmented Generation (RAG), supervised finetuning (FT), Direct Preference Optimization (DPO), and proposed framework (Ours). Accuracy highlighted in green indicates best performance.

Research (2024), OLMo-7B Groeneveld et al. (2024), Gemma 2-27B Team et al. (2024b), Mixtral-8x7B and Mixtral-8x22B Jiang et al. (2024). We also include GPT4o Hurst et al. (2024) and Gemini 1.5 Flash Team et al. (2024a) as representative closed-source models.

## 5.1 SETUP

### 5.1.1 PROMPTING STRATEGIES

We employ an Answer only approach (AO), where the model is given a question with four options and asked to select the correct answer without any explanation relying solely on its pre existing knowledge . In contrast, few-shot prompting Yasunaga et al. (2023) uses a few examples to help the model learn and apply that knowledge to similar tasks. CoT prompting guides the model to generate intermediate reasoning steps, improving its performance on complex tasks by breaking them down into smaller, more manageable parts.

### 5.1.2 RETRIEVAL AUGMENTED GENERATION (RAG)

Unlike vanilla models that rely solely on intrinsic knowledge, our RAG framework using Langchain enhances reasoning by incorporating domain specific context by external physics knowledge PDFs contains formulae only required to solve **?**. PDFs are chunked into 500 character segments with a 50-character overlap to preserve context and embedded using OpenAI's `text-embedding-ada-002`. During inference, the top-k relevant chunks are retrieved and integrated into a structured prompt instructing the model to "Think step-by-step" using the provided context. This approach explicitly injects relevant formulas and concepts and improving solution accuracy in physics reasoning.

### 5.1.3 SLM FINETUNING

We finetuned ultra small models on our finetuning dataset consists of 2,494 high school-level physics questions sourced from standard Indian JEE preparation materials **?**, each paired with a detailed CoT solution. The models include Qwen 2.5 1.5B Instruct, LLaMa 3.2 1B and 3B Instruct, Phi 3.5 mini 3.8B Instruct. finetuning followed a CoT approach, where each input instruction was framed as: *"You are an expert physics assistant. You are given a question. Your task is to generate the final solution of the given question. Let's think step by step."*. We utilized Hugging Face's SFT library with the AdamW optimizer, a learning rate of 5e-5, and a batch size of 2. All models were trained for 3 epochs on a single H100 GPU, with training time ranging from 45 to 60 minutes per model and employed LoRA adapters via the PEFT framework.

### 5.1.4 DIRECT PREFERENCE OPTIMIZATION (DPO)

Starting from our SFT-trained base small LLM model, the pairs consisting of a preferred (correct) and rejected (incorrect) response were used to train the model using the DPO loss, which encourages higher likelihood for preferred responses without requiring on-policy sampling during training. We employed Hugging Face's trl DPOTrainer for this stage. Training was performed on a single H100 GPU using a batch size of 2, gradient accumulation steps of 4, and a learning rate of 5e-5 for 3

epochs. Each training run took approximately 1.5 to 2 hours. The optimizer used was AdamW, and early stopping was employed based on validation loss.

### 5.1.5 EVALUATION

Luo et al. (2023) measure the mathematical reasoning quality of LLMs by directly comparing the final answer and calculating the overall accuracy on a given dataset. We choose to follow the same evaluation. All models are evaluated using final answer accuracy. we further perform a step-by-step comparison when the final answer is incorrect. Using ground truth reasoning traces as reference Xia et al. (2025), we analyze each intermediate step in the model's CoT output to identify specific reasoning error types.

## 6 RESULTS

In Table 1, Ultra small models perform poorly across all benchmarks and prompting strategies, reflecting limited foundational knowledge and a high tendency to hallucinate across all benchmarks. This underscores the complexity of CoT reasoning tasks where models still struggle.

In Table 2 and 3, we evaluate Ultra small model using vanilla CoT reasoning across various prompting strategies. RAG based prompting consistently improved accuracy by providing relevant external context. Surprisingly, FT degraded performance. Analysis suggests that models mimicked the surface form of complex CoT examples without internalizing the underlying logic, leading to flawed intermediate steps and incorrect answers. The fixed solution trajectories in the FT data further limited generalization. Supervised FT alone proved insufficient for teaching robust domain reasoning, especially in smaller models. DPO also failed to improve results, likely because it relies on a competent base model, which in our case lacked reliable intermediate reasoning.

Our approach consistently improves accuracy highlighed in table 2 and 3, across diverse benchmarks and all baselines. Our refinement agents are able to correct early stage errors. Notably, improvements in Ultra small models, suggesting that lightweight, reward guided corrections using agentic feedback can offset capacity limitations. However, in a few cases, performance drops marginally due to over corrections or agent misrouting, especially when initial answers are partially correct but the verifier triggers unnecessary edits. These fluctuations highlight the sensitivity of step level reward modeling and the importance of accurate error localization.

## 7 CONCLUSION

We tackle the underexplored problem of physics reasoning in SLMs, which often fail due to miscomprehension, conceptual, and calculation errors. Existing methods like FT and DPO rely on broad data level alignment but struggle with fine grained, step level corrections. Our core contribution is a lightweight, modular framework that uses agentic feedback and reinforcement learning, guided by a first error step reward. Unlike FT and DPO, our method avoids costly retraining and enables precise, iterative refinement through specialized agents.

## 8 LIMITATION AND FUTURE WORK

One major challenge in building our error localization module was designing a custom verifier to assess SLM generated physics reasoning. Prior studies Ma et al. (2025) emphasize that such verifiers demand large, high quality CoT solutions to perform reliably. Currently, we use LLaMa 3.1 405B as an oracle verifier. Future work includes a lightweight, domain-adapted verifier using a finetuned SLM.

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

# A  APPENDIX

## A.1  PHYSICSQA

### CONSTRUCTION

The PhysicsQA dataset was created to address the gap between foundational knowledge benchmarks and challenging, advanced reasoning problems. It consists of 370 diverse, intermediate-level high school physics problems. These problems were sourced from standard Indian JEE preparation materials from 2000-2010. Each problem is paired with a correct Chain-of-Thought (CoT) solution, which was verified using GPT-4 with human annotators in the loop. This detailed structure allows for a step-by-step evaluation of an LLM's conceptual understanding and calculation abilities, moving beyond simple final answer accuracy. The problems are designed to be challenging, often requiring multiple concepts, intricate calculations, and multi-hop reasoning. A separate fine-tuning dataset, consisting of 2,494 high-school level physics questions also sourced from JEE materials, was curated for the warm-up and RL training stages of our methodology.

The meticulous five-step process used to curate these challenging problems is illustrated in the figure below.

| Chapter Name | Percentage |
|---|---|
| Electromagnetism | 29.8% |
| Mechanics and Kinematics | 21.8% |
| Thermodynamics and Heat | 15.7% |
| Waves and Optics | 15.4% |
| Nuclear and Modern Physics | 8.9% |
| Material Properties and Elasticity | 8.3% |

Table 4: Topic-wise Distribution in PhysicsQA

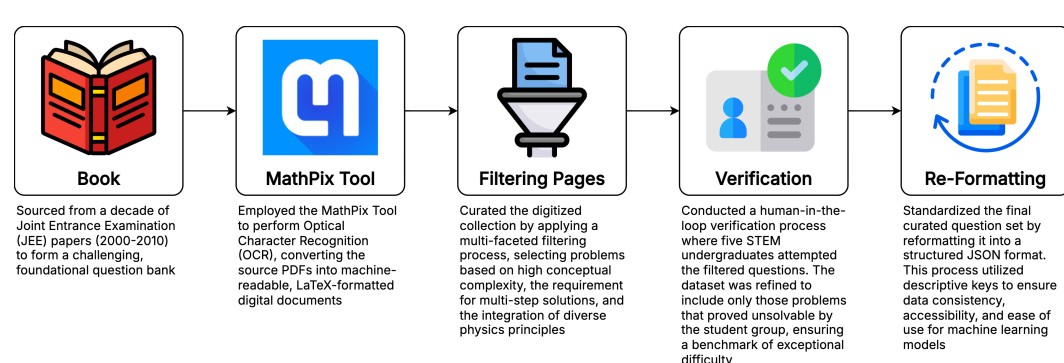

Figure 3: The dataset creation pipeline for PhysicsQA. The process begins with sourcing questions from Joint Entrance Examination (JEE) papers from 2000-2010. These are digitized into LaTeX format using the MathPix OCR tool. A rigorous filtering stage follows, where problems are selected for high conceptual complexity and the need for multi-step solutions. A human-in-the-loop verification process then ensures a high difficulty standard by selecting only those questions that a group of five STEM undergraduates could not solve. Finally, the curated problems are standardized into a structured JSON format with descriptive keys, making them suitable for machine learning applications.

DATASET SAMPLES

**PhysicsQA**

The following figures showcase the diversity of problems within the PhysicsQA dataset, spanning different physics domains and requiring a range of reasoning skills.

"id": "811",

"question": "Prove that the pressure of an ideal gas is numerically equal to two third of the mean translational kinetic energy per unit volume of the gas.",

"cot_solution": "

# Step 1: Understand the problem and identify what needs to be proved. We need to prove that the pressure of an ideal gas is numerically equal to two-thirds of the mean translational kinetic energy per unit volume of the gas.

# Step 2: Recall the formula for translational kinetic energy per unit volume. Translational kinetic energy per unit volume (E) is given by the formula: $E = \frac{1}{2}(\text{mass per unit volume})(\overline{v}^2)$ where $\overline{v}^2$ is the mean squared velocity of the gas molecules.

...

# Step 6: Rearrange the equation to solve for pressure. Rearranging the equation to solve for p, we get: $p = \frac{2}{3}E$. The final answer is: $\left\{ \frac{2}{3}E \right\}$",

"subject": "Physics",

"source": "JEE Questions",

"chapter_name": "Thermodynamics and heat transfer",

"class": "High School"

Figure 4: A sample from the PhysicsQA dataset involving a theoretical proof in thermodynamics. This problem tests the model's ability to reason with abstract concepts and manipulate fundamental formulas, such as the relationship between pressure, density, and kinetic energy, to derive a result rather than calculating a numerical value. It highlights the dataset's inclusion of conceptual reasoning tasks.

"id": "846",

"question": "Example 17: Uranium ores on the earth at the present time typically have a composition consisting of 99.3% of the isotope $\{\}_{92}U^{238}$ and 0.7% of the isotope $\{\}_{92}U^{235}$. The half-lives of these isotopes are $4.47 \times 10^9$ year and $7.04 \times 10^8$ year, respectively. If these isotopes were equally abundant when the earth was formed, estimate the age of the earth.",

"cot_solution": "

# Step 1: Define the initial conditions and the given information.
We are given that uranium ores on the earth at the present time typically have a composition consisting of 99.3% of the isotope $\{\}_{92}U^{238}$ and 0.7% of the isotope $\{\}_{92}U^{235}$. The half-lives of these isotopes are $4.47 \times 10^9$ year and $7.04 \times 10^8$ year respectively.

# Step 2: Determine the decay constants for both isotopes.
The decay constant $\lambda$ is related to the half-life $t_{\frac{1}{2}}$ by the equation $\lambda = \dfrac{0.693}{t_{\frac{1}{2}}}$. Therefore, the decay constants

for the two isotopes are $\lambda_1 = \dfrac{0.693}{4.47 \times 10^9}$ and $\lambda_2 = \dfrac{0.693}{7.04 \times 10^8}$.

...

# Step 6: Substitute the values of the decay constants and calculate the age of the earth.
Substituting the values of the decay constants, we have $t = \dfrac{1}{\frac{0.693}{7.04 \times 10^8} - \frac{0.693}{4.47 \times 10^9}} \ln\left(\dfrac{99.3}{0.7}\right)$. Evaluating this

expression, we get $t = 5.97 \times 10^9$ year.

The final answer is: $\{5.97 \times 10^9\}$",

"subject": "Physics",

"source": "Optics and modern physics",

"chapter_name": "Nuclear physics and Radioactivity",

"class": "12th"

Figure 5: A sample from the nuclear physics and radioactivity chapter. This problem requires multi-step numerical calculation involving half-life, decay constants, and logarithms. It assesses the model's precision with scientific notation and its ability to apply formulas to a real-world scenario (estimating the age of the Earth), demonstrating the dataset's coverage of advanced topics and complex calculations.

**JEEBench**

JEEBench, derived from the Joint Entrance Examination (JEE) Advanced papers in India, represents a benchmark of exceptionally high difficulty, designed to challenge the top echelon of human students. The problems are renowned for requiring deep conceptual understanding, mathematical fluency, and the ability to synthesize multiple topics under pressure.

"description": "JEE Adv 2016 Paper 1",

"index": 2,

"subject": "phy",

"type": "MCQ",

"question": "A uniform wooden stick of mass $1.6\{\sim kg\}$ and length $l$ rests in an inclined manner on a smooth, vertical wall of height $h(< l)$ such that a small portion of the stick extends beyond the wall. The reaction force of the wall on the stick is perpendicular to the stick. The stick makes an angle of $30°$ with the wall and the bottom of the stick is on a rough floor. The reaction of the wall on the stick is equal in magnitude to the reaction of the floor on the stick. The ratio $\dfrac{h}{l}$ and the frictional force $f$ at the bottom of the stick are

$\left(g = 10\{\sim ms\}\{\sim s\}^2\right)$

(A) $\dfrac{h}{l} = \dfrac{\sqrt{3}}{16}, f = \dfrac{16\sqrt{3}}{3}\{\sim N\}$

(B) $\dfrac{h}{l} = \dfrac{3}{16}, f = \dfrac{16\sqrt{3}}{3}\{\sim N\}$

(C) $\dfrac{h}{l} = \dfrac{3\sqrt{3}}{16}, f = \dfrac{8\sqrt{3}}{3}\{\sim N\}$

(D) $\dfrac{h}{l} = \dfrac{3\sqrt{3}}{16}, f = \dfrac{16\sqrt{3}}{3}\{\sim N\}$",

"gold": "D"

Figure 6: A high-complexity problem in static equilibrium. This question requires a model to construct a complete physical model (a free-body diagram), apply both force ($\sum \vec{F} = 0$) and torque ($\sum \vec{\tau} = 0$) equilibrium conditions, and solve a resulting system of simultaneous equations involving trigonometry. It is a rigorous test of translating a physical setup into a solvable mathematical framework.

"description": "JEE Adv 2016 Paper 1",

"index": 12,

"subject": "phy",

"type": "MCQ(multiple)",

"question": "The position vector $\vec{r}$ of a particle of mass $m$ is given by the following equation
$\vec{r}(t) = \alpha t^3 \hat{i} + \beta t^2 \hat{j}$
where $\alpha = \dfrac{10}{3}\{\sim m\}\{\sim s\}^{-3}, \beta = 5\{\sim m\}\{\sim s\}^{-2}$ and $m = 0.1\{\sim kg\}$. At $t = 1\{\sim s\}$, which of the
following statement(s) is(are) true about the particle?
(A) The velocity $\vec{v}$ is given by $\vec{v} = \left(10\hat{i} + 10\hat{j}\right)\{ms\}^{-1}$

(B) The angular momentum $\vec{L}$ with respect to the origin is given by $\vec{L} = -\left(\dfrac{5}{3}\right)\widehat{k}\{\sim N\}\{\sim m\}$

(C) The force $\vec{F}$ is given by $\vec{F} = \left(\hat{i} + 2\hat{j}\right)\{N\}$

(D) The torque $\vec{\tau}$ with respect to the origin is given by $\vec{\tau} = -\left(\dfrac{20}{3}\right)\widehat{k}\{\sim N\}\{\sim m\}$",

"gold": "ABD"

Figure 7: A calculus-intensive vector mechanics problem presented in a multiple-correct format. A model must perform several distinct vector operations—differentiation to find velocity and acceleration, and cross products to find angular momentum ($\vec{L} = \vec{r} \times m\vec{v}$) and torque ($\vec{\tau} = \vec{r} \times \vec{F}$). The multiple-correct nature demands that each option be independently and accurately verified, testing procedural stamina.

"description": "JEE Adv 2017 Paper 2",

"index": 1,

"subject": "phy",

"type": "MCQ",

"question": "Consider an expanding sphere of instantaneous radius $R$ whose total mass remains constant. The expansion is such that the instantaneous density $\rho$ remains uniform throughout the volume. The rate of fractional change in density $\left(\dfrac{1}{\rho}\dfrac{d\rho}{dt}\right)$ is constant. The velocity $v$ of any point on the surface of the expanding sphere is proportional to
[A] $R$
[B] $R^3$
[C] $\dfrac{1}{R}$
[D] $R^{\frac{2}{3}}$",

"gold": "A"

Figure 8: An abstract problem requiring **differential reasoning**. Unlike applying a known formula, this question demands the ability to derive a relationship from first principles. The model must relate density to radius, differentiate their relationship with respect to time, and interpret the result to find the proportionality, testing a model's ability to reason with changing quantities.

COMPARATIVE ANALYSIS: JEEBENCH VS. PHYSICSQA

While JeeBench exemplifies a high level of difficulty in computational and integrative physics, the PhysicsQA dataset is deliberately designed to be a more robust and insightful benchmark for evaluating the reasoning of large language models for several key reasons:

1. **Process vs. Outcome:** JEEBench, with its multiple-choice format, exclusively evaluates the final answer. A correct answer could be achieved through flawed reasoning or even a lucky guess. In contrast, PhysicsQA's structured **chain-of-thought (CoT) solutions** demand the generation of a complete, step-by-step reasoning path. This makes it a superior diagnostic tool, as it assesses the validity of the problem-solving process itself, not just the outcome.

2. **Diversity of Reasoning Tasks:** The JEEBench samples, while complex, are primarily focused on intricate calculation and synthesis within classical mechanics. PhysicsQA demonstrates broader cognitive diversity by including tasks such as **theoretical proofs** (as seen in Figure 4) and **real-world estimation problems** (Figure 5). This provides a more holistic evaluation of scientific intelligence beyond competition-style problems.

3. **Generation vs. Discrimination:** The fundamental task in JeeBench is discriminative selecting the correct option(s) from a provided list. PhysicsQA requires a generative approach creating a detailed solution from scratch. For an AI model, generating a coherent, logically sound, and mathematically correct multi-step solution is a significantly more demanding and authentic measure of true problem-solving ability than choosing an answer.

While the difficulty of JEEBench is undeniable, the architectural design of the PhysicsQA dataset with its focus on the reasoning process, diverse task types, and generative nature—makes it a more powerful and precise instrument for measuring and advancing the scientific reasoning capabilities of large language models.

**SciEval-Static**

The SciEval-Static dataset provides a benchmark for evaluating models on foundational physics problems. The samples are typically multiple-choice questions that require direct application of core principles and formulas, assessing both conceptual knowledge and computational accuracy.

---

"question": "If a 2kg object is constantly accelerated from $0\frac{m}{s}$ to $16\frac{m}{s}$ over 6 s, how much power must be applied at $t = 1$?
     A. 16/3 N
     B. 8/3 N
     C. 32/3 N
     D. 4 N",

"answer": ["A"],

"category": "physics",

"topic": "Work and Energy",

"ability": "Scientific Calculation",

"type": "multiple-choice",

"task_name": "SocraticQA",

"id": "415"

---

Figure 9: A procedural problem from 'Work and Energy' that tests **sequential calculation**. The model must first determine the object's acceleration from kinematic data and then use that result to calculate the force and instantaneous power. This question assesses the ability to follow a clear, step-by-step computational procedure using fundamental formulas ($a = \Delta v/\Delta t$, $F = ma$, $P = Fv$).

"question": "If a concave and convex lens of equal focus are kept in contact, what is the effective focal length?
    A. Infinite focal length (equivalent to plane glass)
    B. Zero focal length
    C. Equal to the original focal length
    D. Half of the original focal length",

"answer": ["A"],

"category": "physics",

"topic": "Refraction",

"ability": "Scientific Calculation",

"type": "multiple-choice",

"task_name": "SocraticQA",

"id": "809"

Figure 10: A **conceptual reasoning** question from optics testing knowledge of the lens combination formula and, critically, scientific sign conventions. The solution hinges on recognizing that a convex $(+f)$ and a concave $(-f)$ lens of equal focal length mathematically negate each other. This problem highlights the dataset's emphasis on testing core physics principles rather than complex arithmetic.

"question": "If a projectile is shot at a velocity of $7\frac{m}{s}$ and an angle of $\frac{\pi}{3}$, how far will the projectile travel before landing?
A. 5.12 m
B. 4.33 m
C. 6.28 m
D. 3.67 m",

"answer": ["B"],

"category": "physics",

"topic": "2D Motion",

"ability": "Scientific Calculation",

"type": "multiple-choice",

"task_name": "SocraticQA",

"id": "97"

Figure 11: A standard projectile motion problem that evaluates a model's ability to perform a **single-step scientific calculation**. Success requires identifying the correct formula for horizontal range $(R = \frac{v_0^2 \sin(2\theta)}{g})$ and accurately substituting the given values. This sample is representative of the dataset's inclusion of foundational, formula-driven problems.

**MMLU-HighSchool**

The MMLU-HighSchool dataset contains questions that primarily evaluate a model's conceptual grasp of physics. The problems often require qualitative reasoning and an understanding of foundational principles rather than intricate multi-step calculations.

"question": "The plates of a capacitor are charged to a potential difference of 5 V. If the capacitance is 2 mF, what is the charge on the positive plate?",

"input": "
    A. 0.005 °C,
    B. 0.01 °C,
    C. 0.02 °C,
    D. 0.5 °C",

"answer": "B",

"topic": "Electromagnetism"

Figure 12: A foundational problem in electromagnetism testing direct **knowledge recall** and application of the core capacitor equation ($Q = CV$). This question also assesses a model's ability to correctly interpret scientific prefixes (mF for milli-Farads), a crucial step in ensuring accurate numerical computation in a straightforward context.

"question": "It is known that a lab cart is moving east at 25 cm/s at time t1 = 0.10 s, and then moving east at 15 cm/s at t2 = 0.20 s. Is this enough information to determine the direction of the net force acting on the cart between t1 and t2?",

"input": "
    A. Yes, since we know the cart is slowing down, its momentum change is opposite the direction of movement, and the net force is in the direction of momentum change.,
    B. No, because we don't know whether forces such as friction or air resistance might be acting on the cart.,
    C. No, because we don't know the mass of the cart.,
    D. Yes, since we know the cart keeps moving to the east, the net force must be in the direction of motion.",

"answer": "A",

"topic": "Mechanics"

Figure 13: A question focused on **qualitative reasoning** in mechanics. It requires no calculation but instead tests a deep conceptual understanding of Newton's Second Law ($\vec{F}_{net} = m\vec{a}$). The model must deduce the direction of the net force from the change in velocity (acceleration), correctly identifying that a decelerating object has a net force opposing its motion.

"question": "A sound wave with frequency f travels through air at speed v. With what speed will a sound wave with frequency 4f travel through the air?",

"input": "A. v/4, B. v, C. 2v, D. 4v",

"answer": "B",

"topic": "Waves"

Figure 14: A **conceptual pitfall** question from the topic of waves. It is designed to test a model's understanding of a core principle: the speed of a wave is determined by the properties of its medium, not its frequency. Success requires resisting the common misconception derived from the wave equation ($v = f\lambda$), highlighting the dataset's role in evaluating robustness against common distractors.

**MMLU-College**

The MMLU-College dataset challenges models with problems that reflect the complexity and interdisciplinary nature of a university-level physics education. These questions often require the synthesis of multiple concepts and a high degree of factual precision.

"question": "The quantum efficiency of a photon detector is 0.1. If 100 photons are sent into the detector, one after the other, the detector will detect photon.",

"input": "
   A. an average of 10 times, with an rms deviation of about 4,
   B. an average of 10 times, with an rms deviation of about 3,
   C. an average of 10 times, with an rms deviation of about 1,
   D. an average of 10 times, with an rms deviation of about 0.1",

"answer": "B",

"topic": "Modern_Physics"

Figure 15: An **interdisciplinary reasoning** problem that merges a concept from modern physics (quantum efficiency) with probability theory. The solution requires not just calculating the expected value but also understanding statistical fluctuations by applying the formula for the standard deviation of a binomial distribution ($\sigma = \sqrt{np(1-p)}$). This sample showcases problems that test the mathematical rigor expected in higher education.

"question": "The coefficient of static friction between a small coin and the surface of a turntable is 0.30. The turntable rotates at 33.3 revolutions per minute. What is the maximum distance from the center of the turntable at which the coin will not slide?",

"input": "
   A. 0.024 m,
   B. 0.048 m,
   C. 0.121 m,
   D. 0.242 m",

"answer": "D",

"topic": "Mechanics"

Figure 16: A classic university-level mechanics problem that requires the **synthesis of multiple concepts**: circular motion (centripetal force) and static friction. A critical component is the **procedural accuracy** needed for unit conversion—transforming revolutions per minute into radians per second. This problem tests a model's ability to construct a solution by integrating different physical principles.

"question": "Electromagnetic radiation provides a means to probe aspects of the physical universe. Which of the following statements regarding radiation spectra is NOT correct?",

"input": "
   A. Lines in the infrared, visible, and ultraviolet regions of the spectrum reveal primarily the nuclear structure of the sample.,
   B. The wavelengths identified in an absorption spectrum of an element are among those in its emission spectrum.,
   C. Absorption spectra can be used to determine which elements are present in distant stars.,
   D. Spectral analysis can be used to identify the composition of galactic dust.",

"answer": "A",

"topic": "Electromagnetism"

Figure 17: A knowledge-intensive question requiring **factual precision** regarding electromagnetic spectra. The 'NOT correct' format tests a model's ability to critically evaluate several statements and identify the specific scientific inaccuracy—in this case, confusing electronic energy level transitions with nuclear structure. This question assesses the breadth and depth of a model's domain-specific knowledge base.

## A.2 RETRIEVAL-AUGMENTED GENERATION (RAG) FOR FORMULAIC ACCURACY

To enhance the model's performance on quantitative problems and mitigate the risk of hallucinating incorrect formulas, we implemented a Retrieval-Augmented Generation (RAG) system. It is crucial to clarify the nature of the retrieval corpus to accurately represent the task posed to the LLM.

Contrary to a system that retrieves from a full textbook, our RAG corpus is intentionally constrained to a concise formula sheet. This knowledge base contains only essential information such as physical constants (e.g., speed of light, Planck's constant), fundamental laws (e.g., Newton's Laws), and topic-specific equations (e.g., for projectile motion or LCR circuits). The full document provided to the RAG system is detailed in the appendix.

This design choice is deliberate. By providing access only to the "tools" of physics the formulas themselves we ensure the model cannot simply find and replicate a pre-existing worked-out solution. The LLM is still required to perform all the critical reasoning steps: understanding the problem, selecting the appropriate formulas from the retrieved context, manipulating them correctly, and executing the final calculation. This approach tests the model's ability to *apply* knowledge, not merely to find and copy it.

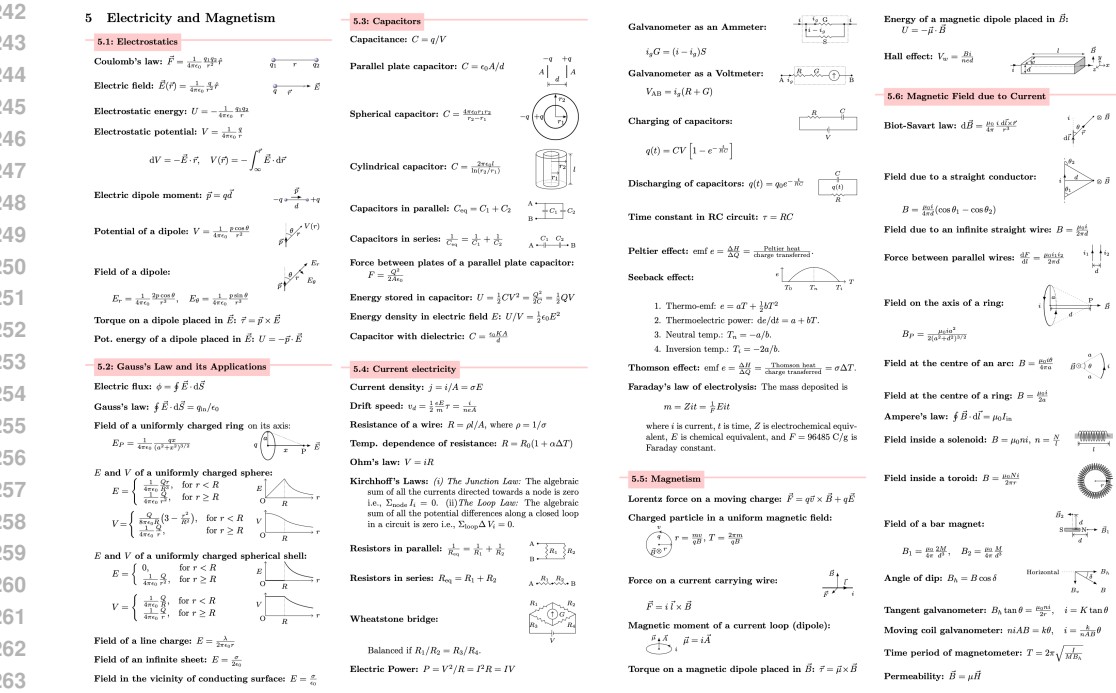

Figure 18: A representative snapshot of the formula sheet used as the knowledge base for our RAG system. The corpus is strictly limited to formulas, definitions, and physical constants, such as Coulomb's Law, capacitor equations, and the Biot-Savart law. The document explicitly **lacks** narrative explanations, problem-solving strategies, or worked-out examples. This design ensures that the RAG acts as a reliable formula reference, compelling the LLM to perform the core reasoning and problem-solving tasks independently.

### A.2.1 Training Dataaset Sample

"id": "721",

"question": "A car moves with a constant tangential acceleration $a_t = \dfrac{2}{3}\dfrac{\text{~m}}{\text{s}^2}$ along a horizontal surface circumscribing a circle of radius $r = 40\text{~m}$. The coefficient of sliding friction between the wheels of the car and the surface is $\mu_s = 0.20$. What distance will the car ride without sliding if at the initial moment of time its velocity is equal to zero ?(take $g = 10\dfrac{\text{~m}}{\text{s}^2}$ )",

"cot_solution": "

# Step 1: Identify the forces acting on the car and their components.
The car is moving with both radial and tangential acceleration, provided by the frictional force. The frictional force $f$ has radial $(f_r)$ and tangential $(f_t)$ components.

# Step 2: Express the radial and tangential components of the frictional force.
$f_r = \dfrac{mv^2}{r}$ and $f_t = ma_t$.

# Step 3: State the condition to avoid sliding.
To avoid sliding, $f = \sqrt{f_r^2 + f_t^2} \leq \mu_s N$, where $N = mg$.

# Step 4: Determine the point of sliding.
At the sliding point, $f_r^2 + f_t^2 = \mu_s^2 m^2 g^2$.

# Step 5: Substitute the expressions for $f_r$ and $f_t$ into the equation for the sliding point.
Substituting for $f_r$ and $f_t$, we get: $V^4 = \left(\mu_s^2 g^2 - a_t^2\right)r^2$.

# Step 6: Use the equation of motion to relate the velocity to the distance traveled.
For zero initial velocity, we have $v^2 = 2a_t S$.

# Step 7: Solve for the distance traveled by the car without sliding.
$\Rightarrow \quad S = \sqrt{\dfrac{\left(\mu_s^2 g^2 - a_t^2\right)r^2}{4a_t^2}} = 40\sqrt{2}\text{~m}.$

The final answer is: $\left\{40\sqrt{2}\right\}$",

"subject": "Physics",

"source": "Physics Module",

"chapter_name": "Mechanics",

"class": "11th"

Figure 19: A sample from our physics Training Dataset, mechanics that requires multi-hop reasoning. The solution involves integrating concepts from dynamics (frictional and centripetal forces) with kinematics to find the final distance. This type of problem tests the model's ability to construct a coherent reasoning chain by combining multiple distinct physics principles, a key challenge in complex problem-solving.

## Preliminary Explanation

### 1. Large Language Models

The rapid advancement of artificial intelligence, particularly with the development of Large Language Models (LLMs) built on the transformer architecture, has redefined the capabilities of natural language processing. These models now exhibit remarkable performance across various language-related tasks, such as text generation, question answering, translation, and summarization, often

rivaling human-like comprehension. More intriguingly, LLMs have demonstrated emergent abilities extending beyond their core functions, showing proficiency in tasks like commonsense reasoning, code generation, and arithmetic. Several key factors have driven the evolution of LLMs, most notably the exponential growth in available data and computational resources. Indeed, on the one hand, social media platforms, digital libraries, and other sources have provided vast amounts of textual and multimedia information, enabling LLMs to be trained on extensive and diverse datasets. On the other hand, the availability of powerful GPUs, TPUs, and distributed computing frameworks has made it feasible to train models with billions, and even trillions, of parameters. Together, these two factors have led LLMs to capture nuanced linguistic patterns, cultural context, and domain-specific knowledge, enhancing their ability to generate coherent, contextually appropriate, and highly versatile outputs.

## 2. PROMPTING TECHNIQUES

**Zero-shot :** Zero-shot prompting refers to a method where a large language model is given a task purely through natural language instructions, without any example inputs or outputs. This technique relies entirely on the model's pretrained knowledge and generalization capabilities. The primary need for zero-shot prompting arises in scenarios where labeled data is unavailable or when rapid deployment is required without fine-tuning. Despite its simplicity, it has proven surprisingly effective in many tasks like classification, summarization, and translation, particularly with larger models like GPT-4. However, its performance can be inconsistent, especially for functions requiring complex reasoning or domain-specific understanding. Furthermore, the outcome is highly sensitive to prompt phrasing, and the lack of examples limits the model's ability to grasp nuanced task expectations.

**Few-shot :** Few-shot prompting builds upon zero-shot by including a small number of input-output examples within the prompt, helping the model better understand the task format and desired response style. This technique addresses the limitations of zero-shot by offering in-context learning, which can significantly enhance performance on structured tasks such as question answering or code generation. It is especially valuable when collecting large training datasets, which are impractical, but a few representative examples are available. Few-shot prompting has demonstrated notable effectiveness, particularly when combined with chain-of-thought examples for reasoning-heavy tasks. Nonetheless, it comes with limitations: the model's performance may degrade if examples are poorly chosen, it has limited capacity to retain many examples due to prompt length constraints, and it still lacks persistent learning across sessions.

**Chain-of-Thought (CoT):** Chain-of-Thought (CoT) prompting is an enhanced strategy developed to augment the performance of large language models (LLMs) on complex reasoning tasks such as arithmetic, commonsense, and symbolic reasoning. This method integrates intermediate reasoning steps within the prompts, providing a more structured path towards the solution. The complexity of reasoning steps is the most critical factor for the performance of LLMs on complex reasoning tasks. Complexity-based prompting can be further enhanced by using the output selection method called Complexity-based Consistency, alleviating the possibility that the model can take shortcuts during reasoning.

## 3. FINE-TUNING [LoRA TECHNIQUE]

In the realm of language models, fine-tuning an existing language model to perform a specific task on specific data is a common practice. This involves adding a task-specific head, if necessary, and updating the weights of the neural network through backpropagation during the training process. It is essential to note the distinction between this fine-tuning process and training from scratch. In the latter scenario, the model's weights are randomly initialized, while in fine-tuning, the weights are already optimized to a certain extent during the pre-training phase. The decision of which weights to optimize or update, and which ones to keep frozen, depends on the chosen technique. Fortunately, parameter-efficient approaches for fine-tuning exist that have proven to be effective. Although most such approaches have yielded less performance, Low Rank Adaptation (LoRA) has bucked this trend by even outperforming full fine-tuning in some cases, as a consequence of avoiding catastrophic

forgetting (a phenomenon that occurs when the knowledge of the pretrained model is lost during the fine-tuning process). LoRA is an improved fine-tuning method. Instead of fine-tuning all the weights that constitute the weight matrix of the pre-trained large language model, two smaller matrices that approximate this larger matrix are fine-tuned. These matrices constitute the LoRA adapter. This fine-tuned adapter is then loaded into the pretrained model and used for inference.

## 4. RAG

Large language models (LLMs) have achieved remarkable success. However, they still face significant limitations, especially in domain-specific or knowledge-intensive tasks, notably producing "hallucinations" when handling queries beyond their training data or requiring current information. To overcome challenges, Retrieval-Augmented Generation (RAG) enhances LLMs by retrieving relevant document chunks from an external knowledge base through semantic similarity calculation. By referencing external knowledge, RAG effectively reduces the problem of generating factually incorrect content. Its integration into LLMs has resulted in widespread adoption, establishing RAG as a key technology in advancing chat-bots and enhancing the suitability of LLMs for real-world applications.RAG research shifted towards providing better information for LLMs to answer more complex and knowledge-intensive tasks during the inference stage, leading to rapid development in RAG studies. As research progressed, the enhancement of RAG was no longer limited to the inference stage but began to incorporate more with LLM fine-tuning techniques. Among the optimization methods for LLMs, RAG is often compared with Fine-tuning (FT) and prompt engineering. Prompt engineering leverages a model's inherent capabilities with minimal necessity for external knowledge and model adaptation. RAG can be likened to providing a model with a tailored textbook for information retrieval, which is ideal for precise information retrieval tasks. In contrast, FT is comparable to a student internalizing knowledge over time, suitable for scenarios requiring replication of specific structures, styles, or formats. RAG excels in dynamic environments by offering real-time knowledge updates and effective utilization of external knowledge sources with high interpretability. However, it comes with higher latency and ethical considerations regarding data retrieval. On the other hand, FT is more static, requiring retraining for updates but enabling deep customization of the model's behavior and style. It demands significant computational resources for dataset preparation and training, and while it can reduce hallucinations, it may face challenges with unfamiliar data. In multiple evaluations of their performance on various knowledge-intensive tasks across different topics, it was revealed that while unsupervised fine-tuning shows some improvement, RAG consistently outperforms it, for both existing knowledge encountered during training and entirely new knowledge. Additionally, it was found that LLMs struggle to learn new factual information through unsupervised fine-tuning. The choice between RAG and FT depends on the specific needs for data dynamics, customization, and computational capabilities in the application context. RAG and FT are not mutually exclusive and can complement each other, enhancing a model's capabilities at different levels. In some instances, their combined use may lead to optimal performance. The optimization process involving RAG and FT may require multiple iterations to achieve satisfactory results.

## 5. RLHF (DPO)

Reinforcement learning from human feedback (RLHF) is a variant of reinforcement learning (RL) that learns from human feedback instead of relying on an engineered reward function. Building on prior work on the related setting of preference-based reinforcement learning (PbRL), it stands at the intersection of artificial intelligence and human-computer interaction. This positioning offers a promising avenue to enhance the performance and adaptability of intelligent systems while also improving the alignment of their objectives with human values. The training of large language models (LLMs) has impressively demonstrated this potential in recent years, where RLHF played a decisive role in directing the model's capabilities toward human objectives. In the RLHF setting, the learning agent needs to solve an RL task without having access to a reward function. To this end, the agent usually simultaneously learns an approximation of the reward function (via the assumed utility function) and an RL policy. Therefore, a generic RLHF algorithm consists of repeating two phases: (1) reward learning and (2) RL training. The first phase can itself be decomposed into two main steps: (i) generate queries to ask the oracle, (ii) train a reward function approximator with the answers provided by the oracle. The RL training part is more conventional and is usually directly based on running a deep RL algorithm using the currently trained reward function approximator.

After learning a reward model, or, more commonly, interleaved with reward model learning, the next step is to train a policy that maximizes the expected accumulated reward. This section will discuss algorithms for policy learning, which can be categorized into two main techniques: adaptation of conventional RL algorithms and direct policy optimization (DPO). Recent studies have also shown that DPO is particularly sensitive to distribution shifts between the base model outputs and the preference data. This sensitivity can lead to poor performance when there's a mismatch between the training data of the base model and the preference dataset. Iterative DPO has been proposed to address this issue. New responses are generated with the latest policy model, and a critique (can be either a separate reward model or the same policy network in a self-rewarding setting) is used for preference labeling in each iteration. This approach can help mitigate the distribution shift problem and potentially improve DPO's performance. Lastly, the tests in the DPO paper were primarily conducted on simple cases, including the IMDB dataset for controlled sentiment generation and the Reddit dataset for summarization. The DPO loss function aimed to maximize the disparity between desired and undesired responses. However, this approach could be problematic. It might lead to simultaneous increases or decreases in the rewards for both desired and undesired responses, as long as the difference between them grows.

## 6. AGENTS

Artificial Intelligence is entering a pivotal era with the emergence of LLM agents, intelligent entities powered by large language models (LLMs) capable of perceiving environments, reasoning about goals, and executing actions. Unlike traditional AI systems that merely respond to user inputs, modern LLM agents actively engage with their environments through continuous learning, reasoning, and adaptation. Compared to conventional agent systems, LLM-based agents have achieved generality across multiple dimensions, including knowledge sources, generalization capabilities, and interaction modalities. LLM agents have found applications across diverse fields, including healthcare, biomedicine, law, and education. LLM agents introduce transformative solutions, such as intelligent online tutoring systems, revolutionizing education accessibility. Planning capabilities are a critical aspect of LLM agents' abilities, enabling them to navigate through complex tasks and problem-solving scenarios with high accuracy. Effective planning is essential for deploying LLM agents in real-world applications, where they must handle a diverse range of complex tasks and scenarios. The planning capability of an LLM agent can be viewed from two perspectives: task decomposition and feedback-driven iteration.

