# OpenReview forum: "Modular Refinement of Small Language Models for Physics Reasoning via Localized Error Feedback"
_ICLR.cc/2026/Conference — Submitted to ICLR 2026_

### Official Review · Reviewer_Zsnf · 2025-10-17

**Soundness:** 1
**Presentation:** 2
**Contribution:** 2
**Rating:** 2
**Confidence:** 4

**Summary:**

The paper first constructs a new physics reasoning benchmark, PhysicsQA, and then proposes to identify errors in reasoning and use reinforcement learning (RL) to refine.
The 3 main error types for physics problem solving  identified by the authors are "Problem Miscomprehension, Incorrect Concept Application, and Calculation Errors."
By designing different modules to refine wrong reasoning steps, the authors leverage RL to tune small language models (SLMs).
Experiments across several physics benchmarks have shown some improvements in the accuracy of SLMs.

**Strengths:**

1. The proposed PhysicsQA dataset might be useful for physics problem-solving with LLMs.
2. The identified three main types of errors in physics reasoning might be useful for further development of stronger LLMs.

**Weaknesses:**

+ This paper brings PhysicsQA, but does not discuss more recent work (early 2025) in constructing benchmarks of physics reasoning.
+ The reward design is questionable. I believe the reward used in the RL process can be easily hacked by just generating fewer steps. For example, the LLM can just put all steps into a "single wrong step".
+ The experiments fail to cover reasoning models. I do not believe this kind of step-level refinement framework can work for reasoning models. Besides, it is hard to segment the whole reasoning process into steps.
+ The method mainly relies on the three main error types. I wonder whether these error types are "complete", i.e., there might be many other error types. This kind of hand-crafted "refinement" module can not generalize to other unseen/undefined error types.
+ The authors mainly focus on physics reasoning. In fact, the current trend treats physics reasoning along with other scientific reasoning/math reasoning as a whole -> "reasoning tasks". Therefore, more benchmarks in other domains should be used.
+ The presentation is poor and confusing:
  + The second paragraph of the introduction is really confusing. It seems like content that should appear in "related work". I can not see any motivation or differences from prior work in the current version of the introduction.
  + The writing needs to be thoroughly improved for the "methodology part". The authors prefer to spend many words on how prior works solve certain challenges. From the writing, it seems all the techniques are already introduced in previous works, and this paper just puts them together.

+ Others: many format issues
  + All subsections are wrongly named, e.g., "2.0.1" -> "2.1"
  + L154-L155: "3"-> "Figure 3"
  + L409 and L418: broken references.
  + Table captions should be put above the table.
  + Table 2 exceeds the margin

**Questions:**

See Weakness

---

### Official Review · Reviewer_3ydd · 2025-10-25

**Soundness:** 2
**Presentation:** 2
**Contribution:** 2
**Rating:** 2
**Confidence:** 4

**Summary:**

This paper proposes a modular refinement framework for physics reasoning in small language models (SLMs). The method performs SFT warm-up followed by an agent-guided RL loop with error localization and targeted feedback. The authors also introduce PhysicsQA, a curated benchmark of 370 high-school-level physics questions with verified CoT traces.

**Strengths:**

Topical relevance — Physics reasoning remains a challenging and important domain for LLMs;

Clear decomposition of failure modes — The taxonomy (miscomprehension / conceptual error / calculation) provides a structured interpretation of LLM errors, aligning method design with observed failure types.

Consistent improvements on small models — On multiple benchmarks, the proposed refinement gives measurable gains over CoT/RAG/FT/DPO on 1B–3B open-source models, showing practical merit under constrained budgets.

**Weaknesses:**

Lack of comparison with existing physics reasoning benchmarks
The paper introduces PhysicsQA as its main benchmark but does not compare or position the proposed method against existing, more challenging physics-reasoning-centered benchmarks, such as Physreason: A comprehensive benchmark towards physics-based reasoning.
Given that Physreason explicitly targets physics reasoning under structured evaluation, the absence of comparison or even discussion makes it unclear whether the proposed framework actually advances physics reasoning, or merely overfits to a custom dataset.

Missing evaluation under strong “deliberate thinking” modes of frontier models
Recent “o-series / R1-style” models (e.g., OpenAI o3-mini(o3-medium), DeepSeek-R1, Gemini-1.5-Pro “thinking mode”) have shown substantial gains on reasoning-heavy tasks simply by changing the inference-time policy (deliberate sampling / longer reasoning / reflection).
The paper does not evaluate how the proposed method compares under identical think-mode inference, nor whether the gains persist if SLM baselines also use deliberation. This omission weakens the claimed contribution.

The “small model deficit” argument is not convincing under current scaling dynamics
The motivation heavily relies on “SLMs struggle, thus SLM refinement is necessary”. But with current and near-term scaling trends (e.g., 7B on consumer hardware with KV-cache offloading, quantization, speculative decoding)，the gap between 3B and 7B inference cost is shrinking rapidly. In this context, the paper does not justify why narrowing the reasoning gap of 3B models is more important than (i) studying scaling laws for physics reasoning, or (ii) using slightly larger, but still affordable, 7B class models.

**Questions:**

See Weaknesses.

---

### Official Review · Reviewer_DGRh · 2025-10-28

**Soundness:** 2
**Presentation:** 1
**Contribution:** 2
**Rating:** 2
**Confidence:** 4

**Summary:**

This paper proposes a method to enhance LLMs' physical reasoning abilities and introduces a new benchmark, PhysicsQA, which contains 370 diverse high school–level physics and math problems.

However, the paper is hard to read, the method is difficult to follow, and there are numerous presentation and formatting issues. In its current form, I do not consider it ready for acceptance at ICLR.

**Strengths:**

**[S1]** The authors construct a manually annotated benchmark, which could be a valuable resource for studying and improving LLMs’ physical reasoning capabilities.

**[S2]** The proposed framework achieves some improvements over the baselines on both JEEBench and PhysicsQA.

**Weaknesses:**

**[W1]** Upon checking the supplementary data, I found issues in data quality. For example, the last test case and the 5th test case are essentially the same, differing only in format. This suggests that the benchmark has not undergone rigorous validation and filtering.

**[W2]** The overall training framework is described vaguely. Despite multiple readings, I found the components difficult to understand, which limits the clarity and impact of the contribution.

**[W3]** Formatting and presentation issues:

+ **[W3.1]** Figure 1 and Table 2 exceed the page margins.

+ **[W3.2]** The images are unclear and should be properly embedded after exporting to PDF.

+ **[W3.3]** Section numbering is incorrect (e.g., Section 2 contains subsubsections labeled as 2.0.1; the same occurs in Section 5).

+ **[W3.4]** Multiple incorrect references appear (e.g., “?” in line 408 and line 418).

+ **[W3.5]** Some reference URLs overflow beyond the page boundary, making them unreadable.

**[W4]** The paper introduces a Feedback Generating Agent powered by Llama 3.1 405B, a very large and capable model. This makes the comparisons in Table 2 potentially unfair, as the strong performance may largely come from this component. The authors should conduct ablation studies to isolate its effect.

**[W5]** The paper lacks any ablation experiments.

**[W6]** As a reinforcement learning–based approach, the method should be compared with other RL baselines such as GRPO to demonstrate its relative advantages.

**[W7]** The proposed method does not clearly exhibit specificity to physical reasoning; it appears applicable to other types of reasoning tasks as well. The authors should explain why the method is particularly suitable for physical reasoning.

**Questions:**

See Weakness

---

### Official Review · Reviewer_Wjy5 · 2025-11-01

**Soundness:** 2
**Presentation:** 3
**Contribution:** 2
**Rating:** 2
**Confidence:** 4

**Summary:**

This paper introduces Agent Guided Feedback training for physics reasoning, along with PhysicsQA, a benchmark of 370 high school physics problems. The authors identify 3 core error types in physics problem solving (for SLMs), and propose to provide step-level guidance via localized, error-specific feedback from an oracle LLM. Experiments show that their method achieves stronger gains compared to other baselines.

**Strengths:**

1. The paper is overall well-motivated.
2. The evaluation covers a wide range of LLMs and benchmarks.
3. The experiments are easy to follow.
4. The curated PhysicsQA benchmark could be a valuable resource to the community

**Weaknesses:**

1. Baseline and ablation for the proposed RL algorithm is insufficient. The authors should at least include a baseline policy gradient algorithm with final answer correctness as the reward, without the agentic feedback loop. If computation resources permit, they should further evaluate their method against mainstream algorithms such as rule-based GRPO or DAPO.
2. Performance analysis for the proposed RL training is missing. The authors should report the throughput or overall GPU hours of each baseline. Since you utilized an LLM feedback loop, I'd expect latency and throughput to be a major concern.
3. Lack of analysis on the feedback quality. Similar to point 1, a simple baseline RL update with no LLM feedback will help quantify the contribution of agent guided feedback. It would be even better if the authors gave some examples of LLM-generated feedback, as well as the reward-training step curve.
4. Lack of detail on experiment setup. For example, what's the context limit for RL training? What's the generation config (e.g. temperature) used for benchmarking?

**Questions:**

1. Why use vanilla policy gradient for RL update? Have you tried mainstream methods such as GRPO?
2. In Fig.2, is there any statistics, on how many internal feedback loop is needed for a given question?
3. Table 2, since you focus on improving physics reasoning of SLMs, why not perform experiments on models with long CoT abilities, such as Qwen3?

---

### Meta-Review · Area_Chair_K5nh · 2025-12-09

**Summary:**

This paper proposes a modular reinforcement learning–based refinement framework to improve physics reasoning in small language models (SLMs). The work combines step-level error localization, targeted feedback from a large oracle LLM, and LoRA-based RLHF to correct three identified failure modes: problem miscomprehension, incorrect concept application, and calculation errors. The authors further introduce PhysicsQA, a benchmark of high-school-level physics problems with verified chain-of-thought traces. Empirical results show the effectiveness of this work.

As there is no rebuttal, and the reviewers' concerns are not addressed, the decision is to reject this paper.

**Reviewer Concerns:**

The reviewers' concerns are as follows.

- Reviewers DGRh, Zsnf, and 3ydd note that the method description is not clear and even verbose, and reads like a composition of existing methods without clear innovation. The modular design is not convincingly differentiated from generic step-wise RLHF or editing frameworks.

- All reviewers highlight the absence of essential comparisons, particularly with standard RL algorithms (e.g., GRPO, PPO) and ablations isolating the contribution of the feedback agent vs. the RL loop itself.

- Reviewer DGRh identifies duplicate questions in PhysicsQA, suggesting insufficient curation. Reviewer 3ydd points out the lack of comparison with established physics reasoning benchmarks like Physreason.

- Reviewer Zsnf argues the step-level reward is easily gameable (e.g., by collapsing reasoning into fewer steps), and the hand-crafted error taxonomy may not generalize to unseen error types.

- Besides, reviewers mentioned severe formatting issues.

**Reviewer Scores:**

Reviewers' ratings are 2/2/2/2, indicating a firm rejection of this paper. As the authors did not provide a rebuttal, the AC concurs with the reviewers.

---

### Decision · Program_Chairs · 2026-01-26

Reject